# Association between preoperative levels of 25-hydroxyvitamin D and hospital-acquired infections after hepatobiliary surgery: A prospective study in a third-level hospital

Estefania Laviano[1]*, Maria Sanchez Rubio[1], Maria Teresa González-Nicolás[1], María Pilar Palacian[2], Javier López[3], Yolanda Gilaberte[4], Pilar Calmarza[5], Antonio Rezusta[2], Alejandro Serrablo[1]

1 Servicio de Cirugía General y Digestiva, Hospital Miguel Servet, Zaragoza, Spain, 2 Servicio de Microbiología, Hospital Miguel Servet, Zaragoza, Spain, 3 Medicina Familiar y Comunitaria, Hospital Miguel Servet, Zaragoza, Spain, 4 Servicio de Dermatología, Hospital Miguel Servet, Zaragoza, Spain, 5 Servicio de Bioquímica, Hospital Miguel Servet, Zaragoza, Spain

* estefania.laviano@gmail.com

**Data Availability Statement:** All relevant data are within the paper.

## Abstract

### Introduction

Evidence implicates vitamin D deficiency in poorer outcomes and increased susceptibility to hospital-acquired infections (HAIs). This study examined the association between serum vitamin D levels and HAIs in a population of hepatobiliary surgery patients.

### Methods

Participants in this prospective analytical observational study were patients who underwent hepatobiliary surgery in a tertiary hospital in Aragon, Spain, between February 2018 and March 2019. Vitamin D concentrations were measured at admission and all nosocomial infections during hospitalization and after discharge were recorded.

### Results

The mean 25-hydroxyvitamin D concentration of the study population (n = 301) was 38.56 nmol/L, which corresponds to vitamin D deficiency. Higher vitamin D concentrations were associated with a decreased likelihood of developing a HAI in general (p = 0.014), and in particularly surgical site infection (p = 0.026). The risk of HAI decreased by 34% with each 26.2-nmol/L increase in serum vitamin D levels.

### Conclusions

Vitamin D levels may constitute a modifiable risk factor for postoperative nosocomial infections in hepatobiliary surgery patients.

**Funding:** None of the author have any financial disclosure. We have not received any external financial aid. Authors did not received any extra money for participating in the study.

**Competing interests:** None of the authors have any conflicts of interests to disclose.

## Introduction

Hospital-acquired infections (HAIs) are a major cause of nosocomial morbidity and mortality. While ongoing prevention programs are indeed necessary, our understanding of the underlying mechanisms that facilitate and ultimately trigger HAIs remains insufficient to effectively address the problem at hand. Existing strategies have primarily focused on attenuating the effects of extrinsic stressors on the host´s immune system (e.g., careful regulation of operating-theatre temperatures, limitation of blood transfusions to reasonable minimums, and shortening surgery times). These strategies seek to modify factors external to the host. While helpful, at best they only enable a modest reduction in postoperative infection rates [1,2].

To our knowledge, few studies have investigated methods that optimize natural host responses to infection in the perioperative setting. One potential point of intervention that has recently attracted attention is the immunological role of vitamin D. Vitamin D receptors are known to be present in most immune system cells, including macrophages, B and T lymphocytes, and neutrophils, and vitamin D regulates the gene expression of antimicrobial peptides as part of the innate immune response [3–7].

A growing body of evidence indicates that vitamin D-deficient patients are more susceptible to nosocomial infections (HAIs) such as pneumonia, urinary tract infections, sepsis, and central line infections [1].

Some studies performed in critically ill patients admitted to an intensive care unit (ICU) suggest that severe vitamin D deficiency before admission is associated with acute kidney injury and mortality [8]. While several reports point to a link between lower vitamin D levels and an increased risk of postoperative HAIs [1–3,9], the influence of vitamin D on surgical outcomes remains unknown. Quaraishi and coworkers reported a significant inverse association between vitamin D levels and HAI risk in bariatric surgery patients [2]. In line with this finding, Turan et al reported an increased risk of HAI and serious complications after noncardiac surgery [3]. Some recent studies have examined the link between surgical site infection (SSI) and preoperative 25-hydroxyvitamin D levels. Abdehgah and coworkers found that vitamin D levels had a strong effect on SSI, but concluded that double-blind trials were needed to confirm this association relationship [10].

The aim of this study was to assess the possible association between serum vitamin D levels and HAIs in patients that had undergone hepatobiliary surgery in a third-level hospital.

## Materials and methods

### Patients and ethics statement

Upon receiving approval from CEICA (The Research Ethics Committee of Aragon, Spain) we performed a prospective analytical observational study of hepatobiliary surgery patients from a third-level hospital in Aragon, Spain. All participating patients were informed in our outpatient clinic about the main study outcomes. CEICA waived the requirement for written informed consent. Patients were free to withdraw consent at any time during the study, although none chose to do so.

Upon agreement to participate in the study, patient data were anonymized by the lead investigator. Neither collaborators nor statisticians had access to non-anonymized data.

Analyses were restricted to patients who underwent hepatobiliopancreatic surgery between February 2018 and March 2019 in the General and Digestive Surgery Service of the Miguel Servet Hospital (Aragon, Spain). Patients who underwent any other type of surgery and those who were admitted but did not undergo surgery due to the presence of unresectable tumours or life-threatening comorbidities were excluded.

### Vitamin D assays

Vitamin D concentrations were measured from serum samples obtained at admission, the day before surgery, and were analysed in our biochemistry laboratory by chemiluminescent microparticle immunoassay (CMIA) using an Alinity i ® Analizer as previously described [7].

CMIA analysis uses paramagnetic microparticles coated with anti-vitamin D antibodies to separate the vitamin from its binding protein. Next, the vitamin D conjugate (vitamin D+ paramagnetic microparticles) is labelled with acridinium to create a chemiluminescent reaction, which is measured in relative light units (RLU). There is an inverse association between the amount of 25-hydroxyvitamin D and the RLU detected by the system [7].

There is some debate regarding optimal vitamin D levels. The Institute of Medicine recommends 25(OH)D levels >20 ng/ml (50 nmol/L), while the American Endocrine Society considers levels >30 ng/ml (75 nmol/L) as optimal, since levels in this range allow maximum calcium absorption from the gut while avoiding hyperparathyroidism [11–15]. Because the Endrocrine Service of our hospital adheres to the guidelines of the American Endocrine Society, we considered levels >30 ng/ml (75 nmol/L) as normal, while levels of 20–30 ng/ml (50–75 nmol/L) and <20 ng/ml (<50 nmol/L) were considered indicative of insufficiency and deficiency, respectively.

### Data collection and statistical analysis

During the course of the study, HAIs acquired by participating patients were documented from admission to discharge and subsequently via scheduled outpatient review appointments. HAIs were diagnosed according to the diagnostic criteria of the Centers for Disease Control and Prevention (CDC) [16]. Cell culture and microorganism identification was performed by hospital's microbiology service.

The information recorded in the study anonymized database included patient demographics, comorbidities, Charlson index score [17,18], type of procedure and duration, use of transfusions and/or vasopressors, ICU admission, length of stay, Clavien Dindo classification [19], reintervention, readmission, and mortality rates.

Categorical variables were expressed as frequency distributions and continuous variables as the median and standard deviation. We used the Shapiro-Wilk test to confirm normal distribution of continuous variables. Distribution was considered normal at p-values >0.05. The Student-Fisher t-test or Mann-Whitney U-test were used for analyses involving 2 continuous quantitative variables, and ANOVA or the Kruskal-Wallis H-test for analyses involving 3 continuous quantitative variables. Categorical variables were analysed using the Chi squared test with a Yates correction. In cases in which the grouping variable had 3 categories, means were compared by ANOVA for continuous variables. Paired (post-hoc) comparisons were performed using Tukey's test for parametric variables or the Benjamini & Hochberg method in cases in which the explanatory variable had a non-normal distribution. Finally, logistic regression analysis was performed to calculate the odds ratio, with the occurrence/absence of the event of interest as the dependent variable and parameters of interest as the independent variable(s). A Wald test was used to calculate the significance of each coefficient in the model. P-values <0.05 were considered statistically significant. In all cases two-sided P-values are reported. Sample size was limited by the recruitment capacity of the hepatobiliary surgery unit. All statistical analyses was performed using R v.3.1.3 statistical software.

## Results

Our study population consisted of 301 patients; 169 men (56.1%) and 132 women (43.9%). The mean 25-hydroxyvitamin D concentration was 38.56 nmol/L, which corresponds to

vitamin D deficiency (vitamin D deficiency, <50 nmol/L; vitamin D insufficiency, 50–75 nmol/L; optimal vitamin D concentration, >75 nmol/L). Table 1 shows summary statistics for baseline demographic data, stratified according to serum vitamin D concentrations and expressed as tertiles. The Charlson comorbidity index is a summative score based on concurrent clinical conditions in a given patient [17].

Vitamin D levels were significantly higher in female than male patients (p = 0.001), accounting for 54% of the third tertile. Mean age was 65.4 years, and vitamin D levels decreased with increasing age (p = 0.002). No significant differences in serum vitamin D levels were associated with the presence of the following comorbidities: diabetes (p = 0.155), chronic obstructive pulmonary disease (COPD) (p = 0.085), cardiovascular disease (p = 0.346), and obesity (p = 0.855). However, we observed a significant association between comorbid hypertension and serum vitamin D levels, which were suboptimal in hypertensive patients. Although most of our patients (49.8%) scored II according to the American Society of Anesthesiology (ASA) patient classification rating, patients with higher ASA (p = 0.002) and Charlson (p = 0.003) scores, had lower serum vitamin D levels.

The most common interventions undergone by patients were cholecystectomy (n = 106, 32%) and minor and major hepatectomies (25.6% and 13.3% respectively). We observed no significant differences in vitamin D levels according to intervention type, although patients that had undergone cholecystectomy had the highest levels of vitamin D. Surgery duration was not associated with any differences in vitamin D levels (p = 0.116).

Patient status according to the Clavien Dindo classification of postoperative complications was associated with significant differences in vitamin D levels (p = 0.031). Patients in the first tertile accounted for almost half of all patients with a Clavien Dindo score >2 points. More than 55% of patients in the third tertile had a Clavien Dindo score of 0 or 1 (Table 1).

Higher vitamin D concentrations were associated with a decreased odds ratio for HAIs in general (p = 0.014), and in particular for surgical site infection (p = 0.026) and central line infection (p = 0.037). For each 26.2 nmol/L increase in vitamin D level the risk of HAI decreased by 34%. Higher vitamin D concentrations were also associated with lower rates of reoperation (p = 0.039), mortality (p = 0.025), and ICU admission (p = 0.005). However we observed no significant association between higher vitamin D concentrations and health care-associated pneumonia (HCAP) (p = 0.089), urinary tract infection (UTI) (p = 0.686), or hospital re-admission (p = 0.828) (Table 2). The estimated common effect odds ratio (OR) for vitamin D levels across individual in-hospital outcomes was 0.67 (CI 0.51–0.88, p = 0.005) for a 26.2 nmol/L increase in vitamin D level.

The estimated odds of developing a HAI decreased almost linearly with increasing vitamin D concentration when all data points were plotted, although 93.02% of patients had vitamin D values <75 nmol/L (Fig 1).

In patients with vitamin D levels <25 nmol/L the odds ratio for HAI was significantly lower than that of patients with vitamin D levels >43.3 nmol/L, but did not differ to that of patients with vitamin D levels within the 25.5–43.3 nmol/L range (Fig 2).

The age, sex, and Charlson index-adjusted model [11,12] (Table 3) revealed that higher vitamin D levels were associated with a 29% reduction (OR 0.71, p = 0.054) in nosocomial infections and a 70% reduction in mortality (OR 0.3, p = 0.056), although these effects were not significant. However, the odds ratio for central line infections was significantly lower in this patient group (OR 0.23, p = 0.023). The model also revealed that the need for transfusions (OR 0.44, p = 0.04) and vasopressors (OR 0.36, p = 0.04) was significantly lower in patients with higher levels of vitamin D. Total in-hospital outcomes were also reduced in this group, with a near insignificant OR of 0.67 (CI 0.51–0.88, p = 0.05).

**Table 1. Demographic data and baseline medical conditions for 301 patients, stratified by tertiles (T) of serum vitamin D concentration.**

| | [Study Population] | T1 [8.8,25.5] | T2 (25.5,43.3] | T3 (43.3,233] | P-value |
|---|---|---|---|---|---|
| N | 301 | 102 | 99 | 100 | |
| Sex: | | | | | <0.001 |
| Males | 169 (56.1%) | 49 (48.0%) | 74 (74.7%) | 46 (46.0%) | |
| Females | 132 (43.9%) | 53 (52.0%) | 25 (25.3%) | 54 (54.0%) | |
| Age | 65.4 (12.7) | 68.0 (11.0) | 66.3 (12.0) | 62.0 (14.2) | 0.002 |
| Diabetes | 61 (20.3%) | 23 (22.5%) | 24 (24.2%) | 14 (14.0%) | 0.155 |
| Hypertension | 151 (50.2%) | 50 (49.0%) | 59 (59.6%) | 42 (42.0%) | 0.044 |
| COPD | 26 (8.64%) | 13 (12.7%) | 9 (9.09%) | 4 (4.00%) | 0.085 |
| Cardiovascular disease | 50 (16.6%) | 21 (20.6%) | 16 (16.2%) | 13 (13.0%) | 0.346 |
| Obesity | 82 (27.2%) | 26 (25.5%) | 27 (27.3%) | 29 (29.0%) | 0.855 |
| ASA index: | | | | | 0.002 |
| 1 | 42 (14.0%) | 9 (8.82%) | 9 (9.09%) | 24 (24.0%) | |
| 2 | 150 (49.8%) | 51 (50.0%) | 47 (47.5%) | 52 (52.0%) | |
| 3–4 | 109 (36.2%) | 42 (41.2%) | 43 (43.4%) | 24 (24.0%) | |
| Charlson Index Score: | | | | | 0.003 |
| 0 | 19 (6.31%) | 4 (3.92%) | 4 (4.04%) | 11 (11.0%) | |
| 1 | 17 (5.65%) | 4 (3.92%) | 2 (2.02%) | 11 (11.0%) | |
| 2 | 15 (4.98%) | 3 (2.94%) | 3 (3.03%) | 9 (9.00%) | |
| 3 | 36 (12.0%) | 13 (12.7%) | 10 (10.1%) | 13 (13.0%) | |
| +4 | 214 (71.1%) | 78 (76.5%) | 80 (80.8%) | 56 (56.0%) | |
| Intervention: | | | | | 0.713 |
| Minor hepatectomy | 77 (25.6%) | 28 (27.5%) | 27 (27.3%) | 22 (22.0%) | |
| Major hepatectomy | 40 (13.3%) | 12 (11.8%) | 12 (12.1%) | 16 (16.0%) | |
| Pancreaticoduodenectomy | 36 (12.0%) | 17 (16.7%) | 12 (12.1%) | 7 (7.00%) | |
| Distal pancreatectomy | 10 (3.32%) | 3 (2.94%) | 3 (3.03%) | 4 (4.00%) | |
| Cholecystectomy | 106 (35.2%) | 32 (31.4%) | 33 (33.3%) | 41 (41.0%) | |
| Exploratory laparotomy | 27 (8.97%) | 9 (8.82%) | 9 (9.09%) | 9 (9.00%) | |
| Others | 5 (1.66%) | 1 (0.98%) | 3 (3.03%) | 1 (1.00%) | |
| Duration | 203 (93.7) | 214 (97.7) | 207 (97.4) | 188 (84.2) | 0.116 |
| Clavien Dindo Classification | | | | | 0.031 |
| 0 | 121 (40.2%) | 37 (36.3%) | 39 (39.4%) | 45 (45.0%) | |
| 1 | 27 (8.97%) | 7 (6.86%) | 5 (5.05%) | 15 (15.0%) | |
| 2 | 50 (16.6%) | 13 (12.7%) | 21 (21.2%) | 16 (16.0%) | |
| 3 | 73 (24.3%) | 28 (27.5%) | 25 (25.3%) | 20 (20.0%) | |
| 4 | 18 (5.98%) | 9 (8.82%) | 6 (6.06%) | 3 (3.00%) | |
| 5 | 12 (3.99%) | 8 (7.84%) | 3 (3.03%) | 1 (1.00%) | |

We divided the sample in tertiles according to vitamin D concentrations so as to obtain a balanced stratification with our reduced sample size. Abbreviations: N,number of patients; T1, first tertile; T2, second tertile; T3, third tertile; COPD, Chronic Obstructive Pulmonary Disease; ASA, American Society of Anesthesiologists.

Multivariate analysis of HAI risk according to vitamin D tertiles adjusted for sex, age, ASA, Charlson index, and surgery type (emergency vs planned) (Table 4) revealed significant differences, with patients in the first tertile at greater risk of acquiring nosocomial infections than those in the third tertile. However, after adjustment for sex, age, transfusions, vasopressors, surgery duration, and length of stay no significant differences were observed between the first and either the second or third tertiles (Table 5).

**Table 2. Associations between serum vitamin D concentration and In-hospital outcomes.**

| In-Hospital Outcome | Incidence N(%) | OR (CI 2.5–97.5%) | P-value (Wald) |
|---|---|---|---|
| HAIs | 91 (28.3) | 0.66 (0.48–0.92) | 0.014 |
| UTI | 6 (1.9) | 0.81 (0.28–2.29) | 0.686 |
| SSI | 83 (25.9) | 0.68 (0.49–0.96) | 0.026 |
| HCAP | 9 (2.8) | 0.33 (0.09–1.18) | 0.089 |
| CAI | 13 (4) | 0.32 (0.11–0.94) | 0.037 |
| ICU | 129 (40.2) | 0.65 (0.49–0.88) | 0.005 |
| Reintervention | 32 (10) | 0.54 (0.3–0.97) | 0.039 |
| Re-admission | 32 (10) | 0.96 (0.63–1.45) | 0.828 |
| Transfusions | 53 (16.9) | 0.45 (0.28–0.75) | 0.002 |
| Vasopressors | 38 (12.1) | 0.34 (0.18–0.65) | 0.001 |
| Mortality | 13 (4.1) | 0.27 (0.09–0.85) | 0.025 |
| In-hospital outcomes | 163 (50.8) | 0.67 (0.51–0.88) | 0.005 |

Logistic regression analysis was performed to determine the odds ratio (OR) for increasing 25(OH)D from the lower quartile (21.5 nmol/L) to the upper quartile (47.7 nmol/L). Abbreviations: CI, confidence interval; CAI, catheter associated bloodstream infections; HAI, hospital-acquired infection; HCAP, health-care associated pneumonia; ICU, intensive care unit; N, number of patients; UTI, urinary tract infection; SSI, surgical site infection.

## Discussion

Vitamin D deficiency is a worldwide pandemic involving both lifestyle- and nutrition-related factors. Even in Spain, where weather conditions could be assumed to facilitate vitamin D metabolism, studies have revealed a paradoxical hypovitaminosis, which cannot be compensated for by mere exposure to sunlight [11,12,20]. The subject of optimal vitamin D serum levels is the focus of an ongoing interdisciplinary debate. The most accepted definition of vitamin D deficiency is a 25-hydroxyvitamin D concentration of less than 50 nmol/L, while insufficiency is diagnosed at concentrations of 50–75 nmol/L [13,15,20,21]. According to these criteria, more than 90% of our study population was either vitamin D deficient or insufficient.

Dividing our sample into tertiles creates a heterogeneous third tertile that includes patients with mild insufficiency and those with optimal vitamin D levels. However, we applied this approach on the basis that our population was insufficiently large to create more categories without losing statistical power.

Recent years have seen an improvement in our understanding of the ways in which vitamin D contributes to vital physiological processes, in addition to its "traditional" role in skeletal metabolism. In their 2012 review of HAIs and vitamin D serum levels, Youssef et al. concluded that patients with vitamin D deficiency had higher rates of infection and that vitamin D levels should be checked upon hospital admission to correct insufficiency [1].

Low vitamin D concentrations are also associated with inflammation, increased risk of cardiovascular disease, and higher insulin resistance and all-cause mortality, especially cardiovascular mortality, given the association between vitamin D deficiency and increased arterial stiffness and endothelial dysfunction in human blood vessels. This association has also been demonstrated in critically ill patients [3,6,7,22–26].

We observed no association between cardiovascular disease and lower vitamin D levels, a consequence of the sample size, study design, and study population. However, we did observe an association between lower vitamin D levels and hypertension (p = 0.044). In line with the fact that patients with low vitamin D serum levels tend to be high-risk patients with multiple comorbidities, ASA (p = 0.002) and Charlson index (p = 0.003) scores were higher in these

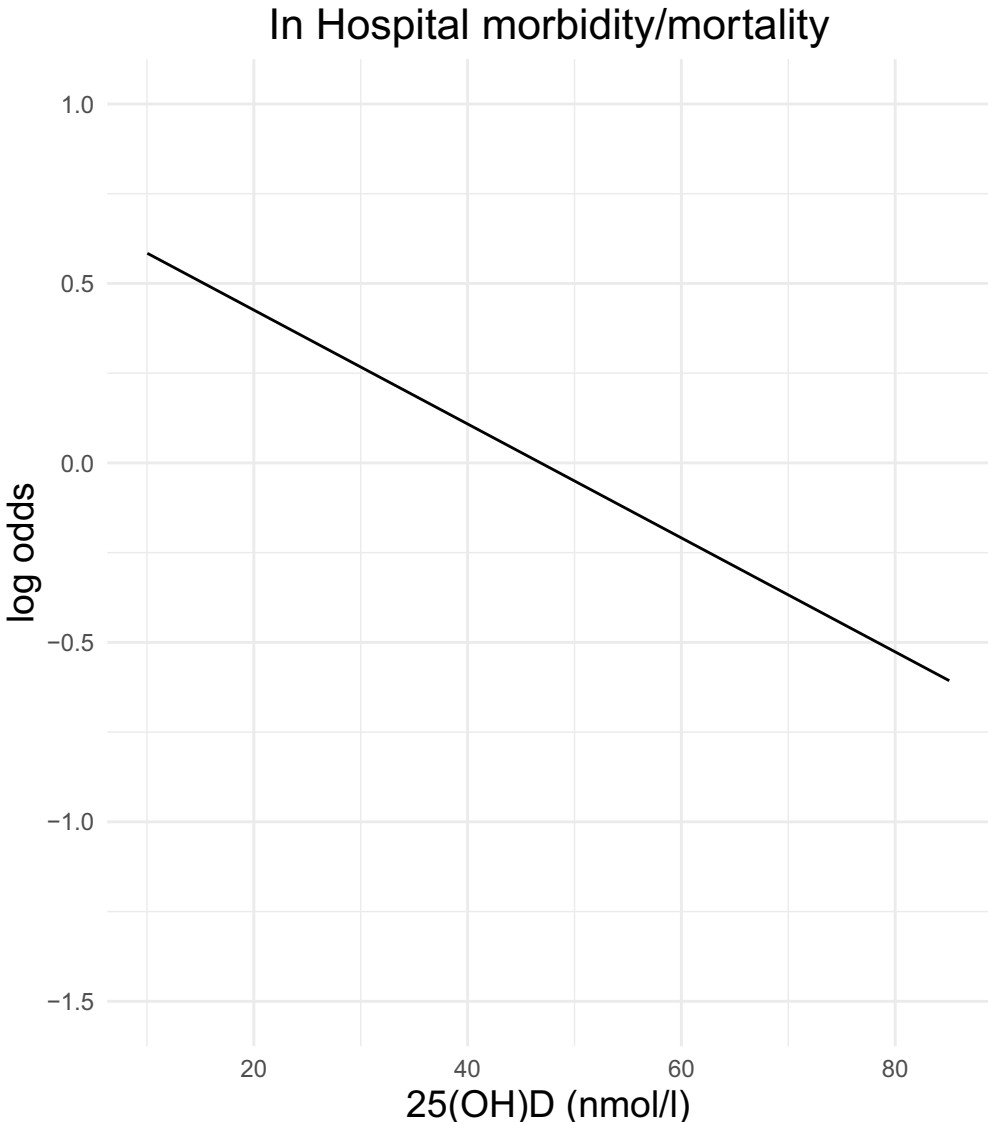

**Fig 1. Risk of in-hospital morbidity/mortality (in log odds) versus vitamin D concentration.** This figure depicts the risk in-hospital morbidity/mortality (log odds) as a function of vitamin D concentration. We observed that the log odds value decreases as vitamin D concentration increases. Probabilities were estimated using a logistic regression model.

patients. We detected no significant associations between vitamin D levels and comorbid diabetes, obesity, or COPD (p>0.005).

Quaraishi and coworkers studied the specific link between vitamin D concentrations and HAIs in surgical patients. In their retrospective analysis of bariatric surgery patients, the authors reported a significant inverse association between preoperative 25 (OH)D levels and the risk of HAI. Several studies have also reported decreased vitamin D bioavailability in obese patients due to sequestration of this fat-soluble vitamin in adipose tissue [2,27,28]. A retrospective study of surgical patients by Turan and colleagues examined the association between serum vitamin D concentration and serious complications after noncardiac surgery. The authors reported an association between vitamin D concentrations and a composite of in-

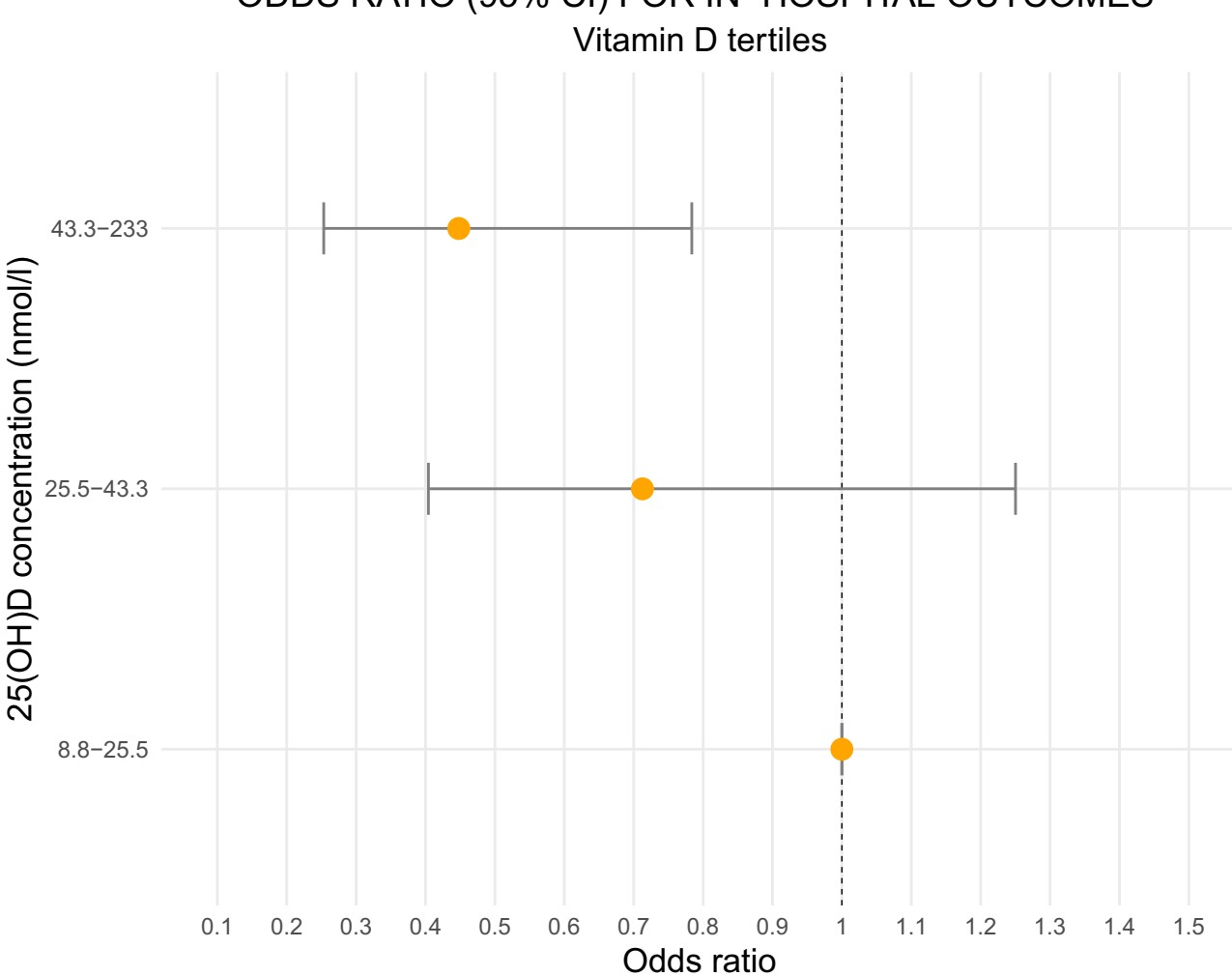

**Fig 2. Raw odds ratios (95% CI) of each tertile of 25(OH)D for in-hospital morbidity/mortality.** Figure shows the odds ratio (yellow dots) and confidence interval (horizontal lines) for in-hospital outcomes for each tertile, using the first tertile as a reference. The odds versus patients with vitamin D <25 nmol/L (reference category) were significantly lower in patients with vitamin D > 43.3 nmol/L, while the odds did not differ significantly in patients with vitamin D 25(OH)D in the range 25.5–43.3 nmol/L.

hospital death, serious infections, and serious cardiovascular events in patients recovering from noncardiac surgery [3].

A search of the literature revealed no studies of vitamin D levels in hepatobiliary surgery patients. Study of this cohort is particularly interesting as it allows comparison of vitamin D levels in cancer patients with those of previously healthy patients, such as those with cholelithiasis [2]. In line with the findings of Turan et al., our results demonstrate a linear inverse association between the risk of HAI and vitamin D serum levels. Moreover, both studies demonstrate an association between higher vitamin D levels and better in-hospital outcomes. However, in both studies multivariate analyses revealed no significant differences in vitamin D levels and HAIs, studied separately, as the effects were attenuated by adjustment variables [3]. Our multivariate analysis of HAI patients as a single group, after adjusting for sex, age, ASA score, Charlson index score, and surgery type (emergency vs planned) (Table 4) revealed a significant decrease in risk between the first and third tertiles. Nonetheless, this effect was not

**Table 3. Associations between serum vitamin D concentration and In-Hospital outcomes.**

| In-Hospital Outcomes | Incidence N(%) | OR (CI 2.5%-97.5%) | P-value (Wald) |
|---|---|---|---|
| HAIs | 91 (28.3) | 0.71 (0.5–1.01) | 0.054 |
| UTI | 6 (1.9) | 0.87 (0.4–1.89) | 0.734 |
| SSI | 83 (25.9) | 0.74 (0.53–1.05) | 0.097 |
| HCAP | 9 (2.8) | 0.36 (0.09–1.38) | 0.135 |
| CAI | 13 (4) | 0.23 (0.06–0.82) | 0.023 |
| ICU admission | 129 (40.2) | 0.71 (0.52–0.98) | 0.04 |
| Reintervention | 32 (10) | 0.6 (0.33–1.1) | 0.098 |
| Re-admission | 32 (10) | 1 (0.65–1.53) | 0.995 |
| Transfusions | 53 (16.9) | 0.44 (0.25–0.78) | 0.004 |
| Vasopressors | 38 (12.1) | 0.36 (0.18–0.73) | 0.004 |
| Mortality | 13 (4.1) | 0.3 (0.09–1.03) | 0.056 |
| In-hospital outcomes | 163 (50.8) | 0.75 (0.56–1) | 0.05 |

Multivariate model adjusted for sex, age and Charlson comorbidity index. Odds ratios (OR) were estimated for increasing 25(OH)D concentrations from the lower quartile (21.5 nmol/L) to the upper quartile (47.7 nmol/L). Abbreviations: CI, confidence interval; CAI, catheter-associated bloodstream infection; HAI, hospital-acquired infection; HCAP, health-care associated pneumonia; ICU, intensive care unit; N, number of patients; UTI, urinary tract infection; SSI, surgical site infection.

**Table 4. Multivariate analysis to estimate the risk (odds ratio) of developing a HAI according to vitamin D concentration, expressed in tertiles and adjusted for sex, age, ASA, Charlson index score, and surgery type (emergency vs planned).**

| | OR | CI (2.5%) | CI (97.5%) | P-value |
|---|---|---|---|---|
| T1 [8.8,25.5nmol/L] | 1 | - | - | - |
| T2 (25.5,43.3 nmol/L] | 0.543 | 0.283 | 1.025 | 0.062 |
| T3 (43.3,233 nmol/L] | 0.476 | 0.236 | 0.94 | 0.035 |

Abbreviations: CI, confidence interval; OR, odds ratio; T1, first tertile; T2, second tertile; T3, third tertile.

**Table 5. Results of multivariate analysis to estimate the risk (odds ratio) of developing a HAI according to vitamin D concentration, expressed in tertiles and adjusted for sex, age, transfusions, vasopressors, surgery duration, and length of stay.**

| | OR | CI (2.5%) | CI (97.5%) | P-value |
|---|---|---|---|---|
| T1 [8.8,25.5nmol/L] | 1 | - | - | - |
| T2 (25.5,43.3 nmol/L] | 0.706 | 0.301 | 1.65 | 0.42 |
| T3 (43.3,233 nmol/L] | 0.908 | 0.398 | 2.091 | 0.818 |

Abbreviations: CI, confidence interval; OR, odds ratio; T1, first tertile; T2, second tertile; T3, third tertile.

observed in the model adjusted for sex, age, transfusions, vasopressors, surgery duration, and length of stay, due to the presence of a large group of patients with benign pathologies (e.g., cholelithiasis) who had a short surgery duration and short length of stay (Table 5).

In their study of patients with health-care associated pneumonia (HCAP), Leow and colleagues [29] demonstrated that severe 25 (OH)D deficiency was common and was associated with a higher 30-day mortality rate than in patients with sufficient 25(OH)D levels during winter. Low vitamin D levels in ICU patients have also been linked to an increased risk of HCAP, although we observed no significant association between these 2 parameters in our study

population [27]. Similarly, we detected no significant association between vitamin D levels and urinary tract infection (UTI), despite the fact that some authors have proposed vitamin D supplementation as a preventive measure for UTI [27]. ICU admission was associated with lower vitamin D levels (p = 0.005), although it should be noted that in our hospital protocol dictates that patients are admitted to ICU after certain interventions, including major hepatectomy and pancreaticoduodenectomy.

Certain limitations of our study should be noted. These include the small sample size, the potentially variable effects of sun exposure on vitamin D levels, and follow-up in outpatient clinic. Although seasonal variations can occur, serum vitamin D levels remain relatively stable over a 1-year period in most patients [30]. However, the mean and median vitamin D levels reported here may differ to those found in other populations and locations, particularly given the potential influence of environmental factors. In our analysis, we sought to account for the effects of multiple confounding variables in order to address this concern. Adjustments to our multivariate analysis were made to account for known confounding variables, including baseline demographics. However, we cannot rule out the existence of unknown factors that could substantially influence our results. Some of our patients received vitamin D supplementation before surgery, and it is unclear how effective these treatments may have been. Obviously, all patients diagnosed with vitamin D deficiency and insufficiency were advised to take supplementation after undergoing surgery.

In summary, our results suggest that preoperative 25(OH)D levels may constitute a modifiable risk factor for postoperative nosocomial infections in hepatobiliary surgery patients. Prospective studies should investigate the potential benefits of optimizing preoperative vitamin D status.

## Acknowledgments

This study was performed at Miguel Servet Hospital, Aragon, Spain without external funding.

## Author Contributions

**Conceptualization:** Estefania Laviano, Yolanda Gilaberte, Pilar Calmarza, Antonio Rezusta, Alejandro Serrablo.

**Data curation:** Estefania Laviano, Maria Sanchez Rubio, Maria Teresa González-Nicolás, María Pilar Palacian, Javier López, Pilar Calmarza, Alejandro Serrablo.

**Formal analysis:** Estefania Laviano, Javier López, Pilar Calmarza.

**Investigation:** Estefania Laviano, María Pilar Palacian, Javier López, Pilar Calmarza, Antonio Rezusta.

**Methodology:** Estefania Laviano, Pilar Calmarza, Antonio Rezusta.

**Project administration:** Yolanda Gilaberte, Antonio Rezusta.

**Resources:** Estefania Laviano, Yolanda Gilaberte.

**Supervision:** Antonio Rezusta, Alejandro Serrablo.

**Validation:** Estefania Laviano.

**Visualization:** Estefania Laviano, Yolanda Gilaberte.

**Writing – original draft:** Estefania Laviano.

**Writing – review & editing:** Estefania Laviano, Alejandro Serrablo.

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
