## [Decision Letter · Decision Letter 0]

21 Feb 2020

PONE-D-20-02256

ASSOCIATION BETWEEN PREOPERATIVE LEVELS OF 25-HYDROXYVITAMIN D AND HOSPITAL-ACQUIRED INFECTIONS AFTER HEPATOBILIARY SURGERY.

PLOS ONE

Dear Dra Laviano,

Thank you for submitting your manuscript to PLOS ONE. After careful consideration, we feel that it has merit but does not fully meet PLOS ONE’s publication criteria as it currently stands. Therefore, we invite you to submit a revised version of the manuscript that addresses the points raised both at the editorial level and a reviewer  comments as noted below during the review process.

We would appreciate receiving your revised manuscript by Apr 06 2020 11:59PM. To enhance the reproducibility of your results, we recommend that if applicable you deposit your laboratory protocols in protocols.io, where a protocol can be assigned its own identifier (DOI) such that it can be cited independently in the future. For instructions see: http://journals.plos.org/plosone/s/submission-guidelines#loc-laboratory-protocols

We look forward to receiving your revised manuscript.

Kind regards,

Dr. Sakamuri V. Reddy

Academic Editor

PLOS ONE

Additional Editor Comments (if provided):

The authors have analyzed a large cohort of 301 patient population to evaluate the association between the levels of 25-OH vitamin D and hospital-acquired infections (HAIs) after hepatobiliary surgery. Specific comments and suggestions to further improve the manuscript are as follow. 1. Abstract- They have noted Discussion at the end, which needs to be replaced with “Conclusions” of the study. 2. (p.6; lines 133-139) delete the ‘title’ of the manuscript here. Also, given the previous studies in the field, the information provided “Introduction” needs to be detailed with more citations. Methods- clarify the statement that the study with patient population is approved with institutional IRB. Also, note separately a section of the “Statistical analysis” of the results. Also, note a separate paragraph on ‘vitamin D’ assays used to consider deficiency as per the American Endocrine Society guidelines (ex., citation #7) in Methods. Results- please provide a clear rationale of the study undertaken given other workers findings on the subject in the field.3. p.18; line 353-Remove citation name “Youssef et al” as it belongs only to citation #20, but it is noted as citations 6,17,20. Figure.1&2 embedded in the text are fine but seems repeated providing eps version again at the end after the References. Provide more details in the legends. I suggest the authors to follow journal article for style.

Journal Requirements:

2. Please provide additional details regarding participant consent. Since the requirement for written informed consent was waived by the ethics committee, please ensure that you have discussed whether all data collected in this study were fully anonymized before you accessed them.

3. In your Methods section, please provide additional information about the participant recruitment method and the demographic details of your participants. Please ensure you have provided sufficient details to replicate the analyses such as: a) the recruitment date range (month and year), b) a description of any inclusion/exclusion criteria that were applied to participant recruitment, c) a description of how participants were recruited, and d) descriptions of where participants were recruited and where the research took place.

4. At this time, we ask that you please provide additional informatino in your Methods section about the methodology used to conduct the chemiluminescent microparticle immunoassay (CMIA) to measure vitamin D concentrations in the patients serum samples.

"NO"

Please provide an amended Funding Statement that declares *all* the funding or sources of support received during this specific study (whether external or internal to your organization) as detailed online in our guide for authors at http://journals.plos.org/plosone/s/submit-now.  Please state what role the funders took in the study.  If any authors received a salary from any of your funders, please state which authors and which funder. If the funders had no role, please state: "The funders had no role in study design, data collection and analysis, decision to publish, or preparation of the manuscript."

6. Thank you for stating the following in your Competing Interests section: 

"NO"

7. Please amend your list of authors on the manuscript to ensure that each author is linked to an affiliation. Authors’ affiliations should reflect the institution where the work was done (if authors moved subsequently, you can also list the new affiliation stating “current affiliation:….” as necessary).

8. Your ethics statement must appear in the Methods section of your manuscript. If your ethics statement is written in any section besides the Methods, please move it to the Methods section and delete it from any other section. Please also ensure that your ethics statement is included in your manuscript, as the ethics section of your online submission will not be published alongside your manuscript.

Reviewers' comments:

Reviewer's Responses to Questions

**Comments to the Author**

1. Is the manuscript technically sound, and do the data support the conclusions?

Reviewer #1: Yes

2. Has the statistical analysis been performed appropriately and rigorously? 

Reviewer #1: Yes

3. Have the authors made all data underlying the findings in their manuscript fully available?

Reviewer #1: Yes

4. Is the manuscript presented in an intelligible fashion and written in standard English?

Reviewer #1: Yes

5. Review Comments to the Author

Reviewer #1: This manuscript by Laviano et al. describes a significant link between preoperative Vitamin-D levels and the increased chance of developing nosocomial infections in patients undergoing hepatobiliary surgery. In general this manuscript is very well-written, easy to follow and its statistical analyses are pretty complete. I would only suggests a few minor additions to the text.

1) In line 161 authors refer previous studies similar to the submitted manuscript. It would be good to have a short explanation stating how this work in different to those.

2) In the legend of the first table the meaning of the abbreviations (ASA, COPD) is not included.

3) Better explanation of how the percentiles in table 1 were calculated is needed.

4) In line 235 authors mentioned Score II, but never mentioned this score in the manuscript. A brief explanation of what this means and why it is relevant to this work will be helpful to follow the manuscript.

5) In the third paragraph of the discussion is mentioned that other studies suggest a link between vit-D levels and heart disease. But this study does not observe statistical significance here. A possible reason for the different results should be included. Also the 4th paragraph of the discussion refers to 3rd paragraph (line 365) so the new paragraph is not really needed.

6. PLOS authors have the option to publish the peer review history of their article (what does this mean?). If published, this will include your full peer review and any attached files.

Reviewer #1: No

---

## [Author Response · Author response to Decision Letter 0]

26 Feb 2020

Reviewer comments:

1. In line 161 authors refer previous studies similar to the submitted manuscript. It would be good to have a short explanation stating how this work in different to those.

A short explanation describing how this study differs to those previously published has been added to the revised manuscript.

2. In the legend of the first table the meaning of the abbreviations (ASA, COPD) is not included.

The legend of Table 1 has been edited to include the meanings of each of the abbreviations used. 

3. Better explanation of how the percentiles in table 1 were calculated is needed.

Further information has been added to describe the process of calculating the percentiles.

4. In line 235 authors mentioned Score II, but never mentioned this score in the manuscript. A brief explanation of what this means and why it is relevant to this work will be helpful to follow the manuscript.

Score II refers to the ASA index score. This information is now included in the methods section.

5. In the third paragraph of the discussion is mentioned that other studies suggest a link between vit-D levels and heart disease. But this study does not observe statistical significance here. A possible reason for the different results should be included. Also the 4th paragraph of the discussion refers to 3rd paragraph (line 365) so the new paragraph is not really needed.

We have corrected the third and the fourth paragraphs of the discussion. We were unable to detect an association between cardiovascular disease and vitamin D levels due to the sample size, sample population, and study design.

---

## [Editor Report · Decision Letter 1]

27 Feb 2020

ASSOCIATION BETWEEN PREOPERATIVE LEVELS OF 25-HYDROXYVITAMIN D AND HOSPITAL-ACQUIRED INFECTIONS AFTER HEPATOBILIARY SURGERY:  A PROSPECTIVE STUDY IN A THIRD-LEVEL HOSPITAL

PONE-D-20-02256R1

Dear Dr. Laviano,

We are pleased to inform you that your manuscript has been judged scientifically suitable for publication and will be formally accepted for publication once it complies with all outstanding technical requirements.

With kind regards,

Sakamuri V. Reddy, Ph.D

Academic Editor

PLOS ONE
---

## [Editor Report · Acceptance letter]

4 Mar 2020

PONE-D-20-02256R1 

Association between preoperative levels of 25-hydroxyvitamin D and hospital-acquired infections after hepatobiliary surgery: A prospective study in a third-level hospital 

Dear Dr. Laviano:

I am pleased to inform you that your manuscript has been deemed suitable for publication in PLOS ONE. Congratulations! Your manuscript is now with our production department. 

With kind regards,

on behalf of

Dr. Sakamuri V. Reddy 

Academic Editor

PLOS ONE